# Observation of a singular Weyl point surrounded by charged nodal walls in PtGa

J.-Z. Ma [1,2,13 ✉], Q.-S. Wu [3,4,13], M. Song [5,6], S.-N. Zhang[3,4], E. B. Guedes[1], S. A. Ekahana[1], M. Krivenkov [7], M. Y. Yao [8], S.-Y. Gao[9], W.-H. Fan[9], T. Qian [9,10], H. Ding [9,10,11], N. C. Plumb [1], M. Radovic [1], J. H. Dil[1,3], Y.-M. Xiong[5], K. Manna [8,12], C. Felser[8], O. V. Yazyev [3,4 ✉] & M. Shi[1 ✉]

Constrained by the Nielsen-Ninomiya no-go theorem, in all so-far experimentally determined Weyl semimetals (WSMs) the Weyl points (WPs) always appear in pairs in the momentum space with no exception. As a consequence, Fermi arcs occur on surfaces which connect the projections of the WPs with opposite chiral charges. However, this situation can be circumvented in the case of unpaired WP, without relevant surface Fermi arc connecting its surface projection, appearing singularly, while its Berry curvature field is absorbed by nontrivial charged nodal walls. Here, combining angle-resolved photoemission spectroscopy with density functional theory calculations, we show experimentally that a singular Weyl point emerges in PtGa at the center of the Brillouin zone (BZ), which is surrounded by closed Weyl nodal walls located at the BZ boundaries and there is no Fermi arc connecting its surface projection. Our results reveal that nontrivial band crossings of different dimensionalities can emerge concomitantly in condensed matter, while their coexistence ensures the net topological charge of different dimensional topological objects to be zero. Our observation extends the applicable range of the original Nielsen-Ninomiya no-go theorem which was derived from zero dimensional paired WPs with opposite chirality.

[1] Photon Science Division, Paul Scherrer Institute, Villigen PSI, Switzerland. [2] Department of Physics, City University of Hong Kong, Kowloon, Hong Kong, China. [3] Institute of Physics, École Polytechnique Fédérale de Lausanne, Lausanne, Switzerland. [4] National Center for Computational Design and Discovery of Novel Materials MARVEL, École Polytechnique Fédérale de Lausanne (EPFL), Lausanne, Switzerland. [5] Anhui Province Key Laboratory of Condensed Matter Physics at Extreme Conditions, High Magnetic Field Laboratory of the Chinese Academy of Sciences, Hefei, Anhui, China. [6] University of Science and Technology of China, Hefei, Anhui, China. [7] Helmholtz-Zentrum Berlin für Materialien und Energie, Elektronenspeicherring BESSY II, Berlin, Germany. [8] Max Planck Institute for Chemical Physics of Solids, Dresden, Germany. [9] Beijing National Laboratory for Condensed Matter Physics and Institute of Physics, Chinese Academy of Sciences, Beijing, China. [10] Songshan Lake Materials Laboratory, Dongguan, Guangdong, China. [11] CAS Center for Excellence in Topological Quantum Computation, University of Chinese Academy of Sciences, Beijing, China. [12] Department of Physics, Indian Institute of Technology Delhi, New Delhi, Delhi, India. [13] These authors contributed equally: J.-Z. Ma, Q.-S. Wu. ✉email: junzhama@cityu.edu.hk; oleg.yazyev@epfl.ch; ming.shi@psi.ch

The Weyl point (WP) in condensed matter is a singular point of Berry curvature that can be regarded as a "magnetic monopole" in momentum space[1]. The electronic band structure in the vicinity of a WP can be described by the $2 \times 2$ Weyl equation[2]. Materials with WPs close to the Fermi level ($E_F$), called Weyl semimetals (WSMs), have attracted significant attention in the recent years[3–10]. It is widely believed that Weyl fermions in any crystal are constrained by the so-called Nielsen–Ninomiya no-go theorem[11,12], which requires that WPs appear in pairs with opposite chirality. A manifestation of the WP pairs is the appearance of surface Fermi arcs connecting the projections of the WPs on the surface Brillouin zone (BZ). Hence, the existence of paired WPs in the bulk and the Fermi arcs on the surfaces has been the essential characteristics to seek out and identify all WSMs to date with no exception. Since the WSM physics has only been developed in the past decade, which is much later than the derivation of the no-go theorem (1980), this theorem does not give any precise description and derivation for its application on the nodal surfaces[11,12]. This kind of topological object and its chirality had not been defined in that year. Recently, it was suggested that the no-go theorem can be circumvented when the WP is surrounded by topological nontrivial Weyl nodal walls (WNWs) and thus a singular WP without opposite chiral point charge counterpart can appear[13]. We extend the original Nielsen–Ninomiya no-go theorem to account for this situation. Namely, instead of seeking for a pair of WPs with opposite chirality, the Berry curvature field of the unpaired WPs can be absorbed by higher dimensional topological objects. In the particular case as mentioned in ref. [13], the Berry curvature of the unpaired WPs are absorbed by the surrounded two-dimensional (2D) WNWs. Therefore, the nodal wall can be viewed as the counterpart of the nodal point. Although it is impossible to choose one closed surface to calculate the Chern number of the WNW on the BZ boundary, we can define the Chern number of the nodal wall from the Berry field theory, i.e., the Chern number of the nodal wall should be opposite to the net charge of all the WPs inside one BZ, which leads to a total zero Chern number in the BZ. Unpaired WP can change the topological properties of WSM, such as well-known chiral anomalous effect, anomalous spin Hall effect, etc. As the singular WP is surrounded by 2D WNWs, no topologically protected Fermi arc surface is necessary. It is considered to be a challenge to realize such kind of WSM in solids with singular unpaired WP near the Fermi level. Except for theoretical predictions[13,14], no unpaired electronic WPs have been observed experimentally so far. Furthermore, electronic topological WNWs/surfaces have not been experimentally reported in crystalline solids up to date. We will show in this work that such kind of Weyl fermions within extended no-go theorem can emerge in a variety of compounds.

The original no-go theorem states that for a general class of fermion theories on a Kogut–Susskind lattice an equal number of species (types) of left- and right-handed Weyl particles (neutrinos) necessarily appears in the continuum limit, by a homotopy theory argument[11,12]. The Weyl particles behave as zero-dimensional chiral nodal points in momentum space. Since the WSM physics is developed only in the recent decade, we can combine recent developed topological physics to simply explain the no-go theorem from a simple perspective in the following. In the three-dimensional (3D) BZ, Chern number can be defined on a 2D slice[3], as shown in Fig. 1a, for a typical WSM constrained by the no-go theorem. Such Chern number can only be defined in the fully gapped 2D slices, because the denominator in the expression of the Chern number becomes zero when there is band degeneracy. The Chern number changes abruptly once the slice passes through a WP. Because the reciprocal lattice is periodic, there must exist another WP with opposite chirality, which restores the Chern number back to its original value when the slice crosses a BZ. A direct consequence of the 2D nonzero Chern number constrained by the no-go theorem is the appearance of surface Fermi arc states, which connect the projections of WPs on the surface BZ[15,16]. The no-go theorem also applies to the recently discovered large Chern number WPs in the CoSi family materials[17–22], i.e., the WPs at high symmetry points are paired and their projections on the surface BZ are connected by Fermi arcs as a consequence of the no-go theorem. As shown in Fig. 1b, the large Fermi arcs connect the projections of the WPs at the center and corners of the surface BZ. The number of Fermi arcs is related to the Chern number on the 2D slices.

Motivated by its fundamental importance, considerable theoretical effort has been dedicated to searching for unpaired WPs. A number of studies have proposed the existence of Weyl fermions beyond the no-go theorem in artificial systems with higher dimensions ($n$-D)[23–25], as well as in non-Hermitian systems[26], in spite of its absence in real applications. Very recently, a theoretical study suggested that Weyl fermions beyond the original no-go theorem can be realized by implementing a single WP surrounded by WNWs[13] (Fig. 1c), a scenario that can be achieved in real crystalline materials (see Supplementary Note 1). In this case, WPs can appear singularly because of the combination of time-reversal and special crystalline symmetry-protected gapless nodal walls on all the BZ boundaries. Due to the existence of the gapless nodal walls, the Chern number in the 2D slices is no longer defined, and thus the WPs do not need to be paired with another point as well as no topologically protected surface Fermi arcs are required. Instead, WP could pair with the 2D WNW. Thus, the no-go theorem could still be applicable when we extend the concept to the case of 2D nodal surface. Despite the appeal of this proposal, it was deemed as a challenging task to realize such a state in the form of a WSM in a condensed matter system. Instead, it was proposed to realize such Weyl fermions in artificial systems, such as cold atoms, photonic/acoustic metamaterials, and circuit networks[13]. Another theoretical report showed that such kind of pairing can appear in some special chiral crystals[14].

## Results

In this work, through symmetry analysis and high-throughput screening based on density functional theory (DFT) calculations, we demonstrate that singular WPs within the extended no-go theorem, which have no associated surface Fermi arc states, can be realized in a series of materials. The considered materials include the PtGa family with space group (SG) No. 198 (Fig. 1e), the $MgAs_4$ family semiconductor (SG No. 92) with a 0.85 eV band gap, a high-pressure phase of elemental Ge (SG No. 96) with a 0.46 eV band gap, $\alpha$-phase $TeO_2$ (SG No. 92), CsSb (SG No. 19), and other compounds for which the computed band structures and topological properties have been retrieved from the TopoMat database[27]. To confirm the theoretical predictions, using angle-resolved photoemission spectroscopy (ARPES), we systematically studied the electronic structure of PtGa, a promising candidate for hosting singular WPs and nontrivial WNWs. We found that, between two bands labeled $N-3$ and $N-2$ (where $N$ is the total number of electrons), there is an ideal singular WP in the center of the BZ, which is enclosed by doubly degenerate WNWs. These topologically charged walls are protected by the time-reversal and nonsymmorphic crystalline symmetries on all the boundaries of the BZ (Fig. 1c, f). Furthermore, we found that there are additional 36 accidental WPs at the generic $\mathbf{k}$ points. In total, the 37 unpaired WPs give a large net positive chiral charge number $+13$ inside the BZ and no surface Fermi arcs associated with these WPs are observed in the ARPES data. The Berry curvature field of the 37 WPs are absorbed by the WNW on the BZ boundary.

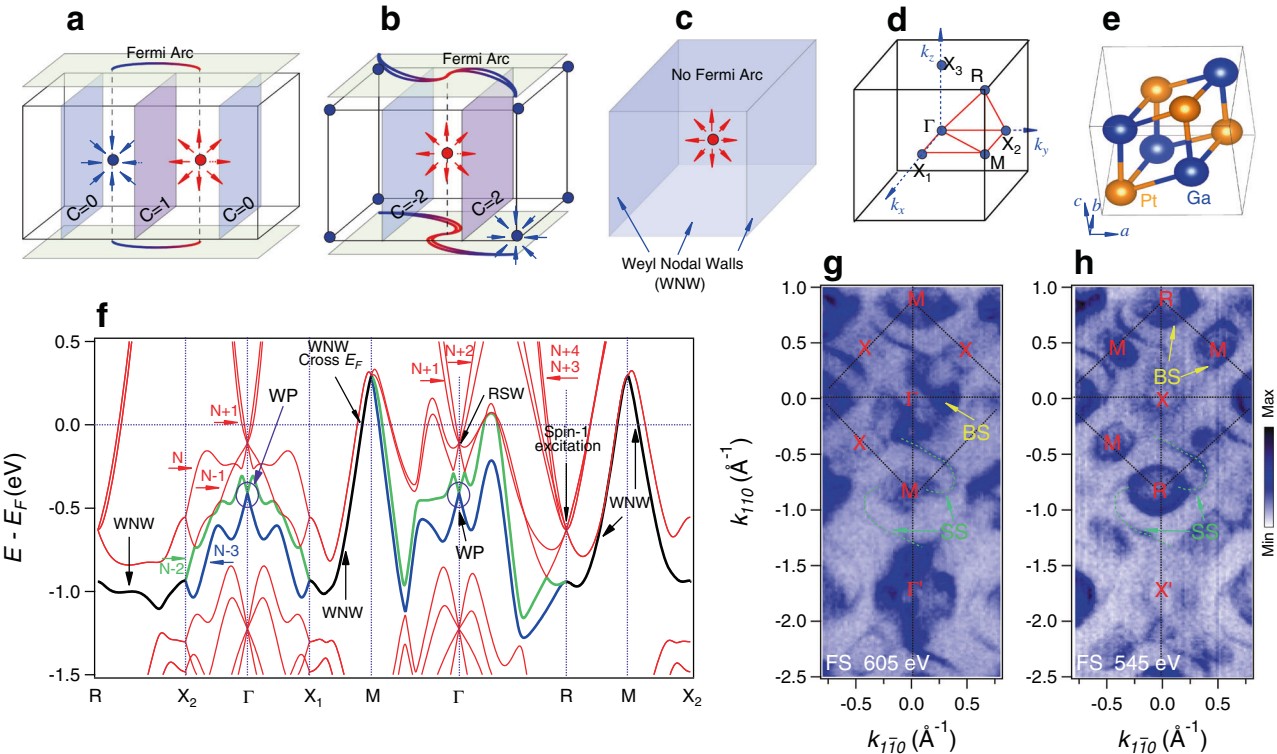

**Fig. 1 Weyl semimetal (WSM) schematic with paired points and unpaired points. a** A conventional WSM restricted by the no-go theorem with surface Fermi arcs surface states connecting the projections of the bulk WPs with opposite chirality on the surface BZ. "C" indicates 2D Chern number on the related slice. **b** The topological semimetal system containing both spin-3/2 RSW (Rarita–Schwinger–Weyl) and spin-1 Weyl points at different high-symmetry points in the bulk BZ and the large surface Fermi arcs connecting the projections of these WPs on the surface BZ. **c** Schematic drawing of a topological semimetal with a single Weyl point in the BZ center enclosed by the topologically charged Weyl nodal walls (WNWs) on the BZ boundaries. No surface Fermi arc connects the projection of this singular WP. **d**, **e** 3D bulk BZ and unit cell of PtGa single crystal, respectively. **f** Bulk band structure of PtGa along the high symmetry lines as indicated in **d**. The green and blue lines are the nondegenerate bands that cross at the BZ boundary forming the topologically charged double degenerate WNWs (black lines). The circles indicate the singular Weyl point in the center of the BZ. **g**, **h** The Fermi surface maps, acquired by ARPES measurements with photon energies $h\nu = 605$ and $545$ eV, respectively. Labels BS and SS denote bulk and surface states, respectively.

**Crystal and electronic structure**. The crystal structure of PtGa belongs to the CoSi family that has been reported to follow the no-go theorem in scenario Fig. 1b. The unconventional Weyl fermions with high chiral charges and large Fermi arcs were discovered in the CoSi family[17–21]. When taking spin–orbit coupling (SOC) into consideration, it is predicted and reported that between bands $N-1$ and $N+4$ the spin-1/2 Weyl fermion, spin-3/2 Rarita–Schwinger–Weyl (RSW) fermion, and double spin-1 Weyl fermion quasiparticles can emerge in the CoSi family[17]. Constrained by the no-go theorem, the RSW node at the BZ center pairs with the spin-1 WP at the $R$ point in the BZ corner, which results in large surface Fermi arcs connecting the projections of the BZ center and corners in the surface BZ (Fig. 1b). However, most of the ARPES measurements were focused on the observation of the large surface Fermi arcs as it was anticipated that the SOC induces only a small energy separation in CoSi; the bulk WPs related to the SOC were not clearly identified[18–22].

As a member of the CoSi family, PtGa has a cubic lattice structure of the space group $P2_13$ (No.198) (Fig. 1e). The corresponding BZ is shown in Fig. 1d. Space group 198 has 12 symmetry operations associated with three basic symmetries: one threefold rotation symmetry along the (111) direction, and two twofold screw symmetry axes along the $z$ and $x$ directions[17], respectively. From the data in the TopoMap database[27] we find that PtGa has the largest SOC effect in the CoSi family, which makes it possible to disentangle the bulk states by using ARPES.

Figure 1g, h show the Fermi surface (FS) maps acquired in the $\Gamma$–$M$–$X$ and $X$–$R$–$M$ planes, respectively. Large Fermi arcs connecting the projections of the $R$ (double Weyl node) and $\Gamma$ (RSW node) points are clearly observed associated with bands $N-1$ to $N+4$. These are indicated by dotted green lines and labeled "SS." The Fermi arcs are furthermore illustrated schematically in Fig. 1b, which have been investigated in the previous report[28]. Meanwhile, the four diamond-like FS patches appear only in the $\Gamma$–$M$–$X$ plane around BZ center (Fig. 1g) but not in $X$–$R$–$M$ plane (Fig. 1h), indicating that these pieces of FS belong to bulk states.

**Singular WP surrounded by WNWs**. Considering the effect of SOC, the band crossing node at the $\Gamma$ point splits into one RSW point and one spin-1/2 WP[17] (Fig. 1f). The spin-1/2 WP is the crossing point of bands $N-2$ and $N-3$ and is protected by time-reversal symmetry. We will show that this is the awaited singular WP whose existence lies within the extended no-go theorem. Preserved by the combination of time-reversal symmetry and nonsymmorphic screw symmetry, the two bands ($N-2$ and $N-3$) are degenerate at all the boundaries of the bulk BZ. In this situation, no surface Fermi arc can be topologically protected relevant to these two bands as the Chern number cannot be defined in the 2D slices of the 3D BZ, which is different to the large surface Fermi arc[28] as discussed above between bands $N-1$ and $N+4$. In Fig. 2b, c, we show the ARPES spectra and the curvature intensity plot along $\Gamma$–$X$ direction. In agreement with

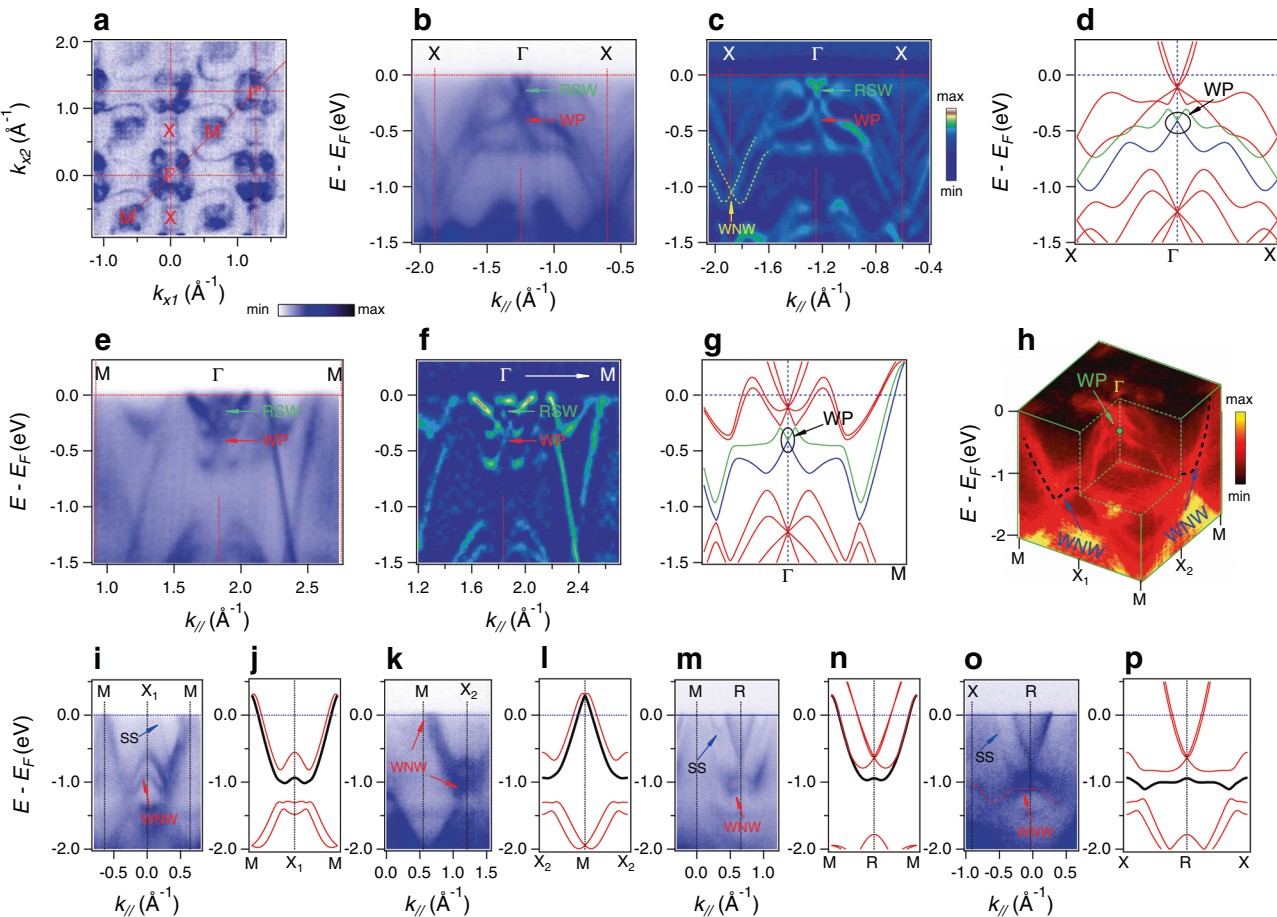

**Fig. 2 Electronic structure of PtGa along the high-symmetry lines showing an singular WP at the center of the BZ surrounded by WNWs on the BZ boundaries. a** FS map in the Γ–X–M plane, acquired with $hv = 605$ eV at 12 K. **b**, **c** High-resolution ARPES spectra along the Γ–X line and the related curvature intensity plot. The red arrows indicate the WP (Weyl point) at the Γ point. **d** Calculated band dispersion along the X–Γ–X line. The singular WP at the Γ point is highlighted by the circle. **e**, **f** The high-resolution ARPES spectra along the Γ–M direction and the related curvature intensity plot. The red arrows indicate the singular WP in the center of the BZ. **g** The calculated band dispersion along the M–Γ–M direction. The singular WP at the Γ point is highlighted by the ellipse. **h** 3D intensity plot of the electronic structure in the Γ–X–M plane; the singular WP at the Γ point and the WNWs (Weyl nodal walls) on the BZ boundary are marked with the filled green circle and black dotted lines, respectively. **i–p** The ARPES spectra and the related calculated band dispersions along high-symmetry lines on the BZ boundaries are displayed alternately. The black dispersive curves represent the WNWs. "SS" denotes surface state.

the calculated band structure (Fig. 2d), two band crossings are observed at the Γ point near the Fermi level, and the SOC-induced band splitting is clearly resolved. The lower band crossing indicated by the red arrows is an unpaired WP, which carries a positive topological charge +1. The related two bands linearly cross at X point forming the WNW, which is clearly resolved in Fig. 2c. To further explore the singular WP, we have also acquired ARPES spectra along the Γ–M direction in the second BZ with $hv = 620$ eV (Fig. 2e, f), which also agree well with the calculated band structure (Fig. 2g). Nevertheless, the WP below the RSW node is clearly observed. An essential condition for the WSM in our scenario is the existence of the symmetry-protected nodal walls on the boundaries of the BZ[13]. In PtGa, the doubly degenerate WNWs on all the BZ boundaries are protected by time-reversal and nonsymmorphic screw symmetries (Fig. 1f). To verify that the WNWs are formed on the BZ boundaries, we collected the ARPES spectra along all the high-symmetry lines on different boundary lines of the bulk BZ (Fig. 2i, k, m, o), i.e., along the $MX_1$, $MX_2$, $MR$, and $XR$ lines in Fig. 1d. The WNWs formed by the crossings of nondegenerate bands $N−2$ and $N−3$ obtained from the band structure calculations (black lines in Fig. 2j, l, n, p) indeed appear in the ARPES spectra, as indicated by the red

arrows in Fig. 2i, k, m, o. The remarkable agreement between the results from band structure calculations and ARPES measurements provides compelling evidence that a single unpaired WP at the center of the BZ is surrounded by the WNWs on the BZ boundary, as it is demonstrated in the 3D plot of the electronic structure in the $k_z = 0$ plane (Fig. 2h).

**Nontrivial FSs.** The nontrivial electronic structure in PtGa associated with bands $N−2$ and $N−3$, which connect the WNW and singular WP, do cross the $E_F$ in the vicinity of the $M$ point. The 3D ARPES intensity plots in the $R–M–X$ and Γ–X–M planes show that band $N−2$ and band $N−3$ cross the Fermi level and form the FS pockets around $M$ points (Fig. 3a, b), which agrees well with the calculated band structure. In our calculations, the WNW spans energies from $−1$ to $0.3$ eV, and the corresponding FSs around the $M$ points are shown in Fig. 3d. To inspect the topological properties of the FS near the $M$ points, we calculated the $x$, $y$, $z$ components of the Berry curvature field on the FS pockets formed by bands $N−3$ (top row of Fig. 3d) and $N−2$ (bottom row of Fig. 3d), respectively. Since the two FS pockets degenerate at the BZ boundary, it is not possible to find a path that enclose only one FS pocket to calculate the Chern number.

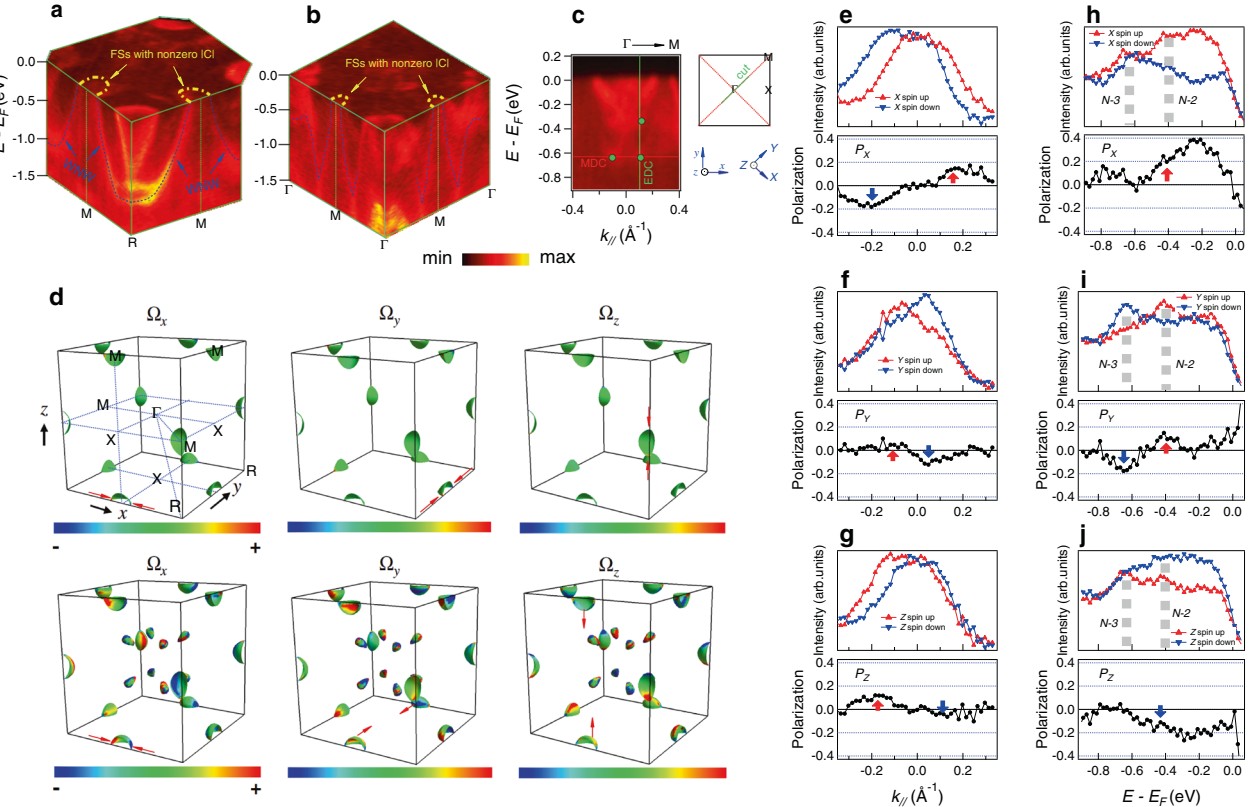

**Fig. 3 Fermi surfaces with nonzero Chern number near $M$ points, and spin polarization of the bands near the singular Weyl point. a, b** 3D band structure along the $R$–$M$–$X$ plane and $\Gamma$–$X$–$M$ plane showing the FS pockets with nonzero Chern number around $M$ points, respectively. The Fermi surfaces are contributed by the cross-section with Fermi level of the two single degenerate bands forming the singular Weyl point and WNW. **c** Spin-integrated band structure along $\Gamma$–$M$ near the singular Weyl points. The MDC and EDC lines indicate the location of the spin measurements. The green dots show the spin-polarized peak position of the bands near the singular WP. **d** The top row shows the Berry curvature field along the $x$, $y$, $z$ directions on the Fermi surface formed by band $N$−3. The bottom row shows the Berry curvature field along the $x$, $y$, $z$ directions on the Fermi surface formed by band $N$−2. **e–g** Measured spin-resolved intensity, along the MDC line in **c**, projected on the $X$, $Y$, $Z$ directions, respectively. The red and blue symbols are the intensity of the spin-up and spin-down states, respectively. The black curves indicate the spin polarization curves. **h–j** Same as **e–g** but along the EDC line in **c**. The spin measurements are taken with circular positive polarized (CP) light source.

However, the topologically nontrivial Chern numbers of these pockets can be unveiled by tracing the flow directions of the Berry curvature. For the FS of band $N$−3, it clearly shows that the Berry curvature field mainly flows into the FS enclosed area from the polar region, indicating that this FS pocket carries nonzero net negative Chern number. On the other hand, for the FS of band $N$−2, the Berry curvature flows both into and out from the enclosed area, which shows dipolar-like topological charges that have been predicted very recently[29]. From a different perspective, the nonzero Chern number from the integration of Berry curvature over the FS (FS Chern number) near the $M$ point should be equal to the net topological charge of the WNW enclosed by the FS. Berry curvature related to nonzero FS Chern number alternates the electronic equation of motion and influences the transport properties, which defines a subject of future studies.

**Nontrivial spin polarization**. Since the band spin splitting induced by inversion symmetry breaking is essential for forming the singular WP in PtGa, we have performed spin-resolved ARPES measurements near the single WP to reveal the spin polarization of the energy bands that form the WP. We focus on the energy distribution curve (EDC) and momentum distribution curve (MDC) along the lines indicated in Fig. 3c in the spectrum along $M$–$\Gamma$–$M$ direction near $\Gamma$ point. The spin-resolved ARPES data were acquired with $h\nu = 95$ eV to probe electronic states in the $\Gamma$–$X$–$M$ plane. Fortunately, as reported in ref. [28] where almost

surface states dominate the intensity in the first BZ, in our measurements the bulk band structure is clearly resolved in the second BZ. Figure 3e–g displays the spin-resolved MDC intensity along the $X$, $Y$, and $Z$ directions (the related coordinate is plotted in Fig. 3c), respectively, which show that the bands are spin polarized along all the $X$, $Y$, and $Z$ directions. The peak positions of spin-polarized spectra are indicated by green dots in Fig. 3c. The data also show that the spin polarization is opposite between the left and right bands centered at the $\Gamma$ point, which obeys both time-reversal symmetry and crystal symmetry, i.e., the spin polarization at any $\mathbf{k}$ point is opposite to that at the $-\mathbf{k}$ point. Furthermore, the $X$, $Y$, $Z$ components of the spin polarized spectra along the EDC is presented in Fig. 3h–j; the change of spin for bands $N$−2 and $N$−3 is clearly resolved. However, due to the absence of mirror symmetry planes and inversion symmetry in the chiral crystal, and in contrast to the well-studied surface states of topological insulators, here all three $P_X$, $P_Y$, and $P_Z$ components are allowed[30] and can show a complex behavior as reflected in the EDC. The spin data in Fig. 3 were acquired with circularly polarized light. To ascertain that the observed spin polarization is intrinsic to the spin-polarized initial states, we repeated the measurements with linearly polarized light, and the obtained spin polarization (Supplementary Fig. 2) is consistent with that shown in Fig. 3. The same spin polarization from differently polarized light gives us the confidence that the observed spin polarizations reflect the intrinsic spin structure of the initial

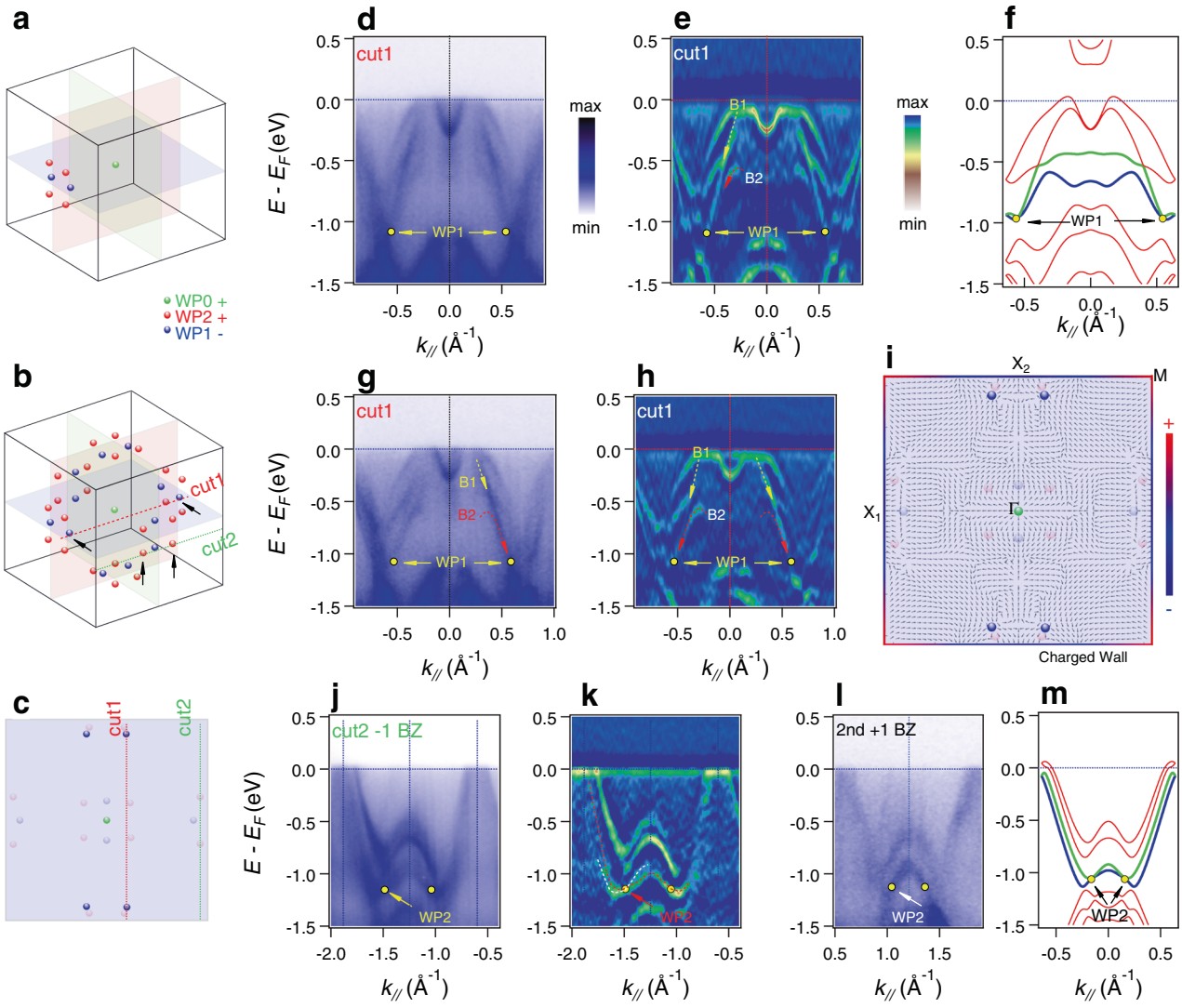

**Fig. 4 Unpaired WPs at generic k points in PtGa. a, b** A group of 6 WPs and their symmetrically equivalent WPs in the **k**-space, respectively. In addition, there is one WP at the center that is discussed in Fig.2, WP0 (green point), with chiral charge +1. There are 12 unpaired accidental WP1 points (blue points) with chiral charge −1 and 24 unpaired accidental WP2 points (red points) with chiral charge +1. The net chiral charge of the WPs in a BZ is +13. **c** Projections of the WPs along the (001), (010), or (100) direction. The WPs marked with opaque (semitransparent) colors are located in (out of) the Γ−X− M plane perpendicular to the projection direction. **d, e** Symmetrized ARPES spectrum along cut1 (in **b**, **c**) across two WP1 and its curvature intensity plot, respectively. The arrows indicate the WP1 points. **f** Calculated band structure along cut1. **g, h** Raw ARPES spectra along cut1 and its curvature intensity plot, respectively. **i** In-plane component of the Berry curvature field in the $k_z = 0$ plane with the projections of the WPs overlaid on top. The colors on the BZ boundaries show the distribution of the topological charge on WNWs. **j, k** The ARPES spectrum along cut2 (in **b**, **c**) across two WP2 points and its curvature intensity plot, respectively. **l** The ARPES spectrum along the cut2 in the second BZ. **m** Calculated band structure along cut2.

state. Thus, the spin polarization of bands $N−2$ and $N−3$ provide additional essential evidence for the existence of the singular WP.

**Accidental WPs at generic k points**. The WNWs on the BZ boundary can absorb all the extra Berry field, allowing unpaired WPs to occur accidentally at generic **k** points inside the BZ[13]. From the calculated results on PtGa, we find that there are two other types of WPs between bands $N−2$ and $N−3$ (WP1 and WP2) at the generic **k** points, i.e., 12 WP1 with negative topological charge −1, and 24 WP2 with positive charge +1 (see Supplementary Table 1). Together with the WP on the Γ point, the net topological charge number of the 37 WPs is positive +13, far beyond zero. The momentum space coordinates, energy, and chirality of the 37 WPs are listed in Supplementary Table 1. To better visualize the locations of the WPs, we divide the 36 WPs into 6 groups. Each group contains two WP1 and four WP2

points near one face of the cubic BZ (Fig. 4a). By applying the twofold rotation (along $k_x$, $k_y$, or $k_z$), threefold rotation (along the 111 direction), and time-reversal operations, we can generate all the positions of the WPs in the whole BZ (Fig. 4b). The ARPES spectrum and its curvature intensity plot along cut1 across two WP1 are shown in Fig. 4g, h. The WP1 points are formed by the crossings of bands B1 and band B2, as marked in the figure. Due to matrix element effects, the intensity of B1 is very weak but still resolvable in the left part of Fig. 4h, while it is more visible in the right part. In contrast, the B2 is clearer in the left side than the right side. To reduce the matrix element effects, we symmetrize the spectrum about the $k = 0$ point (Fig. 4d, e). The two WP1 points become much clearer as indicated by the yellow arrows and found to be consistent with the results of DFT calculations (Fig. 4f). We replot Fig. 4d, e, g, h in Supplementary Fig. 3 with a larger view for a clearer visualization. Figure 4j, k show the

ARPES spectrum and its curvature intensity plot along cut2 across two WP2 points. The spectrum along the same cut in the second BZ is displayed in Fig. 4l. By comparing the data with the calculated band structure (Fig. 4m), we can identify the WP2 points in Fig. 4j–l, as indicated by the arrows. To understand the topological characteristics of the system, we traced the Berry curvature flow between the WNWs and the WPs. As it is difficult to visualize the 3D momentum space, we find it instructive to plot the in-plane component of the Berry curvature in the $k_z = 0$ plane along with the projections of the 37 WPs in Fig. 4i (as well as Supplementary Fig. 4 with a larger view). It can be seen that the distribution of the topological charge on the WNWs is inhomogeneous as some parts of it can be positive (negative) in agreement with the way the Berry curvatures flows out (in) of the WNWs.

## Discussion

Based on our observation and discussion of the novel WPs and WNWs, here we outline the main findings that make PtGa very special and unique compared to normal WSMs: (1) There are 37 WPs in the system between bands $N-3$ and $N-2$. The odd number and large net topological charges of WPs in PtGa indicate that there exist unpaired WPs which are different from all identified 0D WPs that are always paired and possessed zero net charge inside a BZ. (2) First time observation of nontrivial WNWs with large net chiral charges ($-13$/BZ) on the BZ boundary in crystalline solid. Actually, with regard to the source and drain of the Berry curvature field, the singular 0D WPs pair with the 2D WNW; this phenomenon has not been demonstrated experimentally up to date. With this novel observation, we can extend the original Nielsen–Ninomiya no-go theorem from the pairs of WPs to the pairs of topological objects with different dimensions. (3) Except for the large Fermi arcs that connect the surface projections of RSW point and the double spin-1 WP, no other surface states related to the WPs between bands $N-3$ and $N-2$ are observed in the ARPES data, which is consistent with the analysis that there should be no surface Fermi arc connecting the surface projections of the 37 WPs (see Supplementary Fig. 1). (4) All the WP1 (WP2) points in PtGa are equivalent under symmetry operations (twofold rotation along (001) direction, threefold rotation along (111) direction, and time-reversal operations) and have the same chiral charge. The WPs with same sign of charge can disappear when the accidental band crossings at generic **k** points are lifted. But they disappear without annihilating with another opposite WPs, because all the equivalent WPs have the same sign chiral charge, which is, according to one kind accidental band crossings or inversions, either positive or negative. On the other hand, the WP0 at $\Gamma$ point is always protected by the time-reversal symmetry. Thus, when certain types of band inversion disappear, the net charge of all the WPs can change from $+13$ to $-11$, $+25$, or 1 in following the formula $1 + 24 \times m - 12 \times n$, where $m$ and $n$ are either 0 or 1. Therefore, although the net topological charge of the WPs inside the BZ (excluding the nodal wall) can be different from $+13$, it is always a nonzero odd number because of the special protected WP at $\Gamma$ point. These findings extend our understanding of the topological nature of Weyl fermions in condensed matter.

From theoretical considerations and band structure calculations, we further predict: (1) Such novel WPs can also be realized in all members of the CoSi family, including CoSi, AlPt, RhSi, FeSi, ReSi, and other materials. A particularly interesting compound is ReSi because a single unpaired WP at the $\Gamma$ point is almost exactly at the Fermi level (Supplementary Fig. 5), which makes it an ideal candidate for studying transport properties of

this special WSM; (2) Various other families of materials, e.g., a high-pressure phase of Ge (Supplementary Fig. 6), the MgAs$_4$ semiconductor family (Supplementary Fig. 7), $\alpha$-phase TeO$_2$ (Supplementary Fig. 8), etc., are also promising candidates for hosting singular Weyl fermions. Without topological surface states, the response to external field will directly indicate the bulk Weyl fermion characteristics for these materials. Moreover, magnetotransport phenomena such as the chiral anomaly and anomalous Hall effect, which are related to the geometry of the locations of the paired WPs, are thought to be the fingerprints of WSM. Meanwhile, the geometry between unpaired WPs and WNWs is quite different from that of normal WSMs, so that the transport properties are expected to change significantly in this new kind of WSMs, which needs further study in the future. The Weyl fermion pairs between 0D points and 2D nodal wall in crystalline solids will open a new avenue to the bulk topological properties of Weyl fermions in solids, which will promote the understanding of basic topological physics, and application of WSMs into spintronics.

## Methods

**Crystal synthesis**. Single crystals of PtGa were grown using the self-flux technique. First, the stoichiometric polycrystalline sample was melted at 1150 °C, halted for 10 h, and then slowly cooled to 1050 °C with a rate of 1 °C/h. In order to improve the sample quality, the grown crystal was further annealed at 850 °C for 120 h and then slowly cooled to 500 °C at a rate of 5 °C/h. The single crystallinity was checked using a room temperature white-beam backscattering Laue X-ray set-up as well as a single-crystal X-ray diffractometer.

**ARPES measurements**. ARPES measurements were performed at ADRESS-ARPES beamline and SIS beamline of the Swiss Light Source (PSI), at the DREAMLINE of Shanghai Synchrotron Radiation Facility (SSRF), and at the beamlines UE112 PGM-2b-1^3 and UE112 PGM-2a-1^2 at BESSY (Berlin Electron Storage Ring Society for Synchrotron Radiation) synchrotron. The energy and angular resolutions were set to 5–100 meV and 0.1°, respectively. The SARPES measurements were performed at the COPHEE end station of the Swiss Light Source with an energy (angular) resolution of 60 meV (1.5°). The polished sample for (S)ARPES measurements were treated with sputter-annealing method in the vacuum with temperature up to 700 °C and measured in a temperature range between 10 and 20 K in a vacuum better than $2 \times 10^{-10}$ Torr.

**First-principles calculations**. The first-principles calculations have been performed using the Vienna ab initio simulation package (VASP)[31,32] within the GGA approximation. The cutoff energy of 520 eV was chosen for the plane wave basis. The lattice constants and atomic positions have been adopted from the ICSD database without carrying out any additional relaxation. The location of WPs and the calculation of their chirality have been performed using the open-source code WannierTools[33] based on the Wannier tight-binding model constructed using the Wannier90 code[34].

## Data availability

Data that support the conclusions of this study are available in a public repository (MARVEL Materials Cloud Archive) with the same title of this paper (https://archive.materialscloud.org).

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

## Acknowledgements

We acknowledge T. L. Yu, V. N. Strocov, E. Rienks, A. Varykhalov, and Y. B. Huang for help during the ARPES experiments. This work was supported by the NCCR-MARVEL funded by the Swiss National Science Foundation, the Sino-Swiss Science and Technology Cooperation (Grant No. IZLCZ2-170075), and the European Union's Horizon 2020 research and innovation program under the Marie Skłodowska-Curie grant agreement No. 701647. M.R. and J.-Z.M. were supported by the project 200021_182695 funded by the Swiss National Science Foundation. J.-Z.M is supported by City University of Hong Kong through the start-up project (Project No. 9610489). K.M., M.Y.Y., and C.F. acknowledges financial support from European Research Council Advanced Grant No. (742068) "TOP-MAT," European Union's Horizon 2020 research and innovation program (grant No. 824123 and 766566) and Deutsche Forschungsgemeinschaft (Project-ID 258499086 and FE 633/30-1). K.M. acknowledges Max Plank Society for the funding support under Max Plank–India partner group project. H.D. and T.Q. acknowledge financial support from the Ministry of Science and Technology of China (2016YFA0401000, and 2016YFA0300600), the National Natural Science Foundation of China (U1832202), and the Chinese Academy of Sciences (QYZDB-SSW-SLH043, XDB33000000, and XDB28000000.).

## Author contributions

M. Shi and J.-Z.M. supervised this project. J.-Z.M. performed normal ARPES experiments with help from S.A.E., M.Y.Y., S.-Y.G., W.-H.F., M.R., M.K., T.Q., and H.D.; J.-Z.M. performed and analyzed the SARPES experiments with help from E.B.G. and J.H.D.; J.-Z.M. analyzed the ARPES data and plotted the figures. J.-Z.M. searched the database and found all the candidate compounds, which are confirmed by Q.-S.W. via calculations; Q.-S.W., S.-N.Z., and O.V.Y. performed first-principles calculations of the band structure. K.M. and C.F. synthesized and polished the single crystals of PtGa. M. Song and Y.-M.X. synthesized single crystal of PtGa for the primary ARPES study. All the authors contributed to the discussions. J.-Z.M. and M. Shi wrote the manuscript with help from S.A.E., N.C.P., J.H.D., and O.V.Y.

## Competing interests

The authors declare no competing interests.
