## [Peer Review File · Nature Communications]

REVIEWER COMMENTS

Reviewer #1 (Remarks to the Author):

The manuscript by J.-Z. Ma et al. reported the experimental observation of Weyl fermions in PtGa, including multifold chiral fermions with giant Fermi arcs and other two-fold Weyl points. Normal two-fold Weyl has been observed in Ref. [7,8] for the first time. Multifold chiral fermions and their giant Fermi arcs have been highlighted by reference [17-19]. The main contributions of the authors are the observation of two-fold Weyl pinned at Γ point, which is interesting itself.

However, the authors are following a susceptible concept "beyond the Nielsen-Niomiya no-go theorem". The concept of a pinned two-fold Weyl Fermions at Γ and a paired charged nodal surface/wall was first proposed by Nature Materials 17, 978-985 (Fig. 3, and SI F. Topology of Nodal Surfaces in Nonsymmorphic Chiral Space Groups). The materials provided by the authors, PtGa (SG 198), MgAs₄ (SG 92), Ge (SG 96), and CsSb (SG 19), all within the space groups listed in Nature Materials 17, 978-985 (Table II). According to that work, the nodal wall/surface contains the chiral charge to pair with the Weyl Fermions at Γ . In other words, the net charge is always 0 in the whole BZ, which of-course preserves the Nielsen-Ninomiya no-go theorem, where left- and right-handed Weyl-like chiral degeneracies appear in pairs.

The authors' experimental evidence of "beyond the Nielsen-Niomiya no-go theorem" is mainly based on the absence of Fermi arcs. However, this is not surprising when the bands are not very clean, with many projected bulk states to cover surface states. Actually, lacking clear Fermi arcs is the main experimental challenge in demonstrating Weyl Fermions in ARPES. Therefore, under current data shown in figs. 3 and 4, the authors have not even provided concrete experimental evidence of the Chern number of the points observed.

Therefore I can not recommend the publication of this work in the current format.

Reviewer #2 (Remarks to the Author):

The manuscript by Ma et al. presented ARPES measurements on PtGa and reported the observation of single Weyl point beyond the Nielsen-Ninomiya no-go theorem. The findings are novel; The measurements and analyses are thorough. Therefore, I support the publication if the authors can address the following comments

1. What is the photon energy for the spin-resolved measurement? Is it also in the soft x-ray regime? Are the authors should the photon energy used for spin measurement is sensitive to the bulk bands?
2. There is already an ARPES paper on PtGa published in Nature Commun (Yao et al. Nature Communications 11, 2033 (2020)). Surprisingly, this paper was not cited. The authors may cite that paper. Moreover, they should clearly explain the similarity and difference between these two papers.

Reviewer #3 (Remarks to the Author):

Using ARPES and DFT calculations, authors have proposed and provided first experimental evidence on a new type of Weyl semimetals that have unpaired Weyl points. The main finding is interesting and most of the ARPES data are of high quality, therefore it is suitable for publication in Nat. Commus. I have some comments and suggests:

1. N-2 and N-3 bands always degenerated at BZ boundaries, therefore Chern number cannot be defined in the 2D slice. I suggest authors to add some simple explanation on this issue for general readers.

2. Line 190, Fig. 3D should be Fig. 3d. line 214, Fig. 3E-G should be Fig. 3e-g; Line 254, "white arrows", in Fig. 4, is it "yellow arrows"?

3. In the calculations, there are 36 accidental WPs. In figure 4, authors tried to show two of them. However, those ARPES spectra are not clear at all. Therefore, the existence of those WPs is mainly based on the DFT calculations. How reliable is the calculation on these 36 WPs? Are there any effects that could possibly remove the degeneracy of the crossing point to reduce the number of WPs? If so, will the net chiral charge number possibly reduce to zero? Is there any symmetry protection that can always keep the total chiral charge of the real materials none zero?

We would like to thank the Referees for their efforts in reviewing our manuscript and for their appreciation of the significance of our work. The constructive and thoughtful questions, comments and suggestions were very helpful for improving the manuscript. In this response, we address all the issues raised by the Referees point-by-point and indicate the corresponding changes made in the revised manuscript in the last page of this reply letter.

~~~~~  
**Reply to Reviewer #1**  
~~~~~

We sincerely thank the Referee for his/her evaluation, stating that “The main contributions of the authors are the observation of two-fold Weyl pinned at Γ point, which is interesting itself.”

Q: However, the authors are following a susceptible concept “beyond the Nielsen-Niomiya no-go theorem”. The concept of a pinned two-fold Weyl Fermions at Γ and a paired charged nodal surface/wall was first proposed by Nature Materials 17, 978-985 (Fig. 3, and SI F. Topology of Nodal Surfaces in Nonsymmorphic Chiral Space Groups). The materials provided by the authors, PtGa (SG 198), MgAs₄ (SG 92), Ge (SG 96), and CsSb (SG 19), all within the space groups listed in Nature Materials 17, 978-985 (Table II). According to that work, the nodal wall/surface contains the chiral charge to pair with the Weyl Fermions at Γ . In other words, the net charge is always 0 in the whole BZ, which of-course preserves the Nielsen-Ninomiya no-go theorem, where left- and right-handed Weyl-like chiral degeneracies appear in pairs.

A: We would like to thank the Referee for raising this point.

Firstly, we share the same view point with the referee that “...the observation of two-fold Weyl Fermions pinned at Γ point, which is interesting itself.” In the reference (Nature Materials 17, 978-985) mentioned by the reviewer, the authors study chiral crystals and listed all the 65 space groups with chiral crystal structure. Thus, it is not surprising that their 65 space groups include what we mentioned (4 space groups) in our manuscript, but it does not mean that all the chiral space groups host such kind of singular Weyl point. Instead, only several (6 discovered till now) space groups could host singular Weyl points. The authors devoted only a very limited section to discuss such situation through DFT calculations without discussion the relationship with the no-go theorem, without description of surface states, and with no experimental observations. Up to date, the two-dimensional (2D) Weyl nodal surface/sphere/wall has not been

experimentally demonstrated, let alone the pairing between 0D Weyl point and 2D Weyl nodes. In our manuscript we report the experimental observation of such kind of topological objects and their connection for the first time, and the overlap between our manuscript and the Nature Material paper is very little. We have cited that paper and alleviated our calculation description in the reversed manuscript.

Secondly, it is impossible to choose one enclosed path to calculate the topological charge of the Weyl nodal wall. Only under the premise that the net charge of the BZ is zero, we obtain the topological charge of the Weyl nodal wall, i.e., opposite to the net charge of the 37 Weyl nodes inside the whole BZ. We stated in our manuscript that all the Berry field of the Weyl points will be absorbed by the Weyl nodal wall, i.e., all the singular Weyl points pair with the Weyl nodal wall. However, if we consider merely the 0D nodes (Weyl points) as described in the two original Nielsen-Niomiya no-go theorem papers, e.g. Weyl points in all the experimentally determined WSMs, the Weyl points themselves in PtGa are not fully paired to yield a nonzero net chiral charge.

Thirdly, the Nielsen-Ninomiya no-go theorem states that for a general class of fermion theories on a Kogut-Susskind lattice an equal number of species (types) of left- and right-handed Weyl particles (neutrinos) necessarily appears in the continuum limit, by a homotopy theory argument. Since the Weyl semimetal physics is developed only in recent decade, which is much later than the time at which the no-go theorem (1980) is derived, this theorem doesn't give any precise description and derivation for its application on the nodal surfaces. This kind of topological object and its chirality had not been defined in that years. This is the reason why we use the word "beyond", i.e. our observation is beyond the prerequisites of the original no-go theorem. However, if one generalizes the no-go theorem to include different-dimensional topological objects, the extended no-go theorem is applicable for special cases of the concomitant coexistence of singular Weyl nodes and separated nodal lines or nodal walls. In some other cases, it is still not applicable as shown in Ref. [arXiv:2104.00294v1 (2021)]. Moreover, the scenario of topological surface states will be very different to the cases with only 0D Weyl points.

To make this point clearer, we have added more description about the concept in the reversed manuscript about how to extend the no-go theorem in the case containing the 2D Weyl nodes. We changed the title. We replaced all description about "beyond Nielsen-Ninomiya no-go theorem" with a more accurate "extended no-go theorem" as discussed above.

Q The authors' experimental evidence of "beyond the Nielsen-Niomiya no-go theorem" is mainly based on the absence of Fermi arcs. However, this is not surprising when the bands are not very clean, with many projected bulk states to cover surface states. Actually, lacking clear Fermi arcs is the main experimental challenge in demonstrating Weyl Fermions in ARPES. Therefore, under current data shown in figs. 3 and 4, the authors have not even provided concrete experimental evidence of the Chern number of the points observed.

A: We respectfully disagree with the referee on this point. In the manuscript we have explained that the 2D Chern number in the 2D slices perpendicular to the surface (as shown in Fig. 1a-c) leads to the appearance of Fermi arc being ill-defined. Lack of topological protection and definition, the appearance of Fermi arcs is no longer necessary. In such a case, one can no longer count the number of Fermi arcs to confirm the Chern number of the Weyl point. Our calculation and experimental observation both proved the nonexistence of the surface Fermi arc. Thus, it is impossible and not reasonable for us to provide the evidence of surface Fermi arc that does not exist. Instead, we have provided the spin texture data near the Weyl point in Fig.3. Considering the Weyl point at BZ center is formed by two spin nondegenerate bands which linearly cross at one point from all directions in momentum space shown in Fig.2, we already experimentally got the conclusion that the value of the Chern number is $|C|=1$.

~~~~~  
**Reply to Reviewer #2**  
~~~~~

We thank the Referee for pointing out that our results are novel by stating "The findings are novel; The measurements and analyses are thorough. Therefore, I support the publication if the authors can address the following comments". We are indebted to him/her for the valuable suggestions, which were helpful for improving the manuscript.

Q: 1. What is the photon energy for the spin-resolved measurement? Is it also in the soft x-ray regime? Are the authors should the photon energy used for spin measurement is sensitive to the bulk bands?

A: The referee raised a very good question. The photon energy for the spin data is recorded at 95eV which is related to the Γ XM plane that agrees with the inner potential get from our photon energy dependent experiments. Actually, for PtGa, photoemission in both UV and soft X-ray regime are sensitive to the bulk bands. However, in the UV regime the ARPES signal in the 1st BZ is dominated by the surface states associated to the RSW, as reported in the [Nature Comm 11, 2033 (2020)], thus one is not able to observe the bulk states clearly in 1st BZ. It's true that soft x-ray is more sensitive for the bulk band and that's why we used soft x-ray to prove the single Weyl point with high k_z momentum resolution. However, for spin-ARPES measurements the resolution is usually much worse in terms of both momentum and energy, the advantage of high k_z momentum resolution using soft x-ray is strongly diminished, considering that there are rare facilities in the world hosting soft x-ray spin-ARPES beamline. Fortunately, in UV regime we could resolve the bulk band clearly in the second BZ, as the surface bands are suppressed by matrix element effect. We have added more description about the photon energy information in the reversed manuscript.

Q: 2. There is already an ARPES paper on PtGa published in Nature Commun (Yao et al. Nature Communications 11, 2033 (2020)). Surprisingly, this paper was not cited. The authors may cite that paper. Moreover, they should clearly explain the similarity and difference between these two papers.

A: We thank the referee for pointing out the previous paper on PtGa which is published by our co-authors. We apologise for not citing it, as we believe that there is very little overlap between that paper and this manuscript. Now we have cited it in the reversed manuscript and explained the difference between the two papers.

In the Nature Comm 11, 2033 (2020) paper, the authors focus on the surface Fermi arcs connecting surface projections of the RSW node at the BZ center and the double Weyl nodes at the BZ corner, which deals with the topology between bands N-1 to N+4. We also observed and explained such kind of Fermi arcs in our manuscript as shown in the main texts and Fig. 1b,g,h.

Our manuscript mainly deals with the bulk unpaired Weyl points and the Weyl nodal wall between bands N-2 and N-3. Therefore, the two works deal with completely different band indices and different topology. Just by accident, the two topological band structures appear in the same material due to the special chiral symmetries.

~~~~~  
**To Reviewer #3**  
~~~~~

We thank the Referee for his/her appreciation of the significance of our work by pointing out “Using ARPES and DFT calculations, authors have proposed and provided first experimental evidence on a new type of Weyl semimetals that have unpaired Weyl points. The main finding is interesting and most of the ARPES data are of high quality, therefore it is suitable for publication in Nat. Commus.”.

Q: 1.N-2 and N-3 bands always degenerated at BZ boundaries, therefore Chern number cannot be defined in the 2D slice. I suggest authors to add some simple explanation on this issue for general readers.

A: We thank the referee for the suggestions. We have added further explanations regarding this issue in the reversed manuscript. When we define the Chern number on a 2D slice, the related bands should be nondegenerate on all points on the slice. This is because, in the formula of the Chern number as shown in the following, the denominator is zero when there is band degeneracy between bands m and n. Thus, one can only define and calculate 2D Chern number on the gapped slices.

$$C = \frac{1}{2\pi} \int d\hat{S} \cdot \hat{\Omega}_n$$
$$\hat{\Omega}_n = i \sum_{m \neq n} \frac{\langle n | \nabla_k \hat{H} | m \rangle \times \langle m | \nabla_k \hat{H} | n \rangle}{(\varepsilon_m - \varepsilon_n)^2}$$

where $\varepsilon_m, \varepsilon_n$ are the energies of the bands with index m and n.

Q: 2.Line 190, Fig. 3D should be Fig. 3d. line 214, Fig. 3E-G should be Fig. 3e-g;Line 254, “white arrows”, in Fig. 4, is it “yellow arrows”?

A: We would like to thank the referee, we corrected all these typos in the reversed manuscript.

Q: 3.In the calculations, there are 36 accidental WPs. In figure 4, authors tried to show two of them. However, those ARPES spectra are not clear at all. Therefore, the existence of those WPs is

mainly based on the DFT calculations. How reliable is the calculation on these 36 WPs? Are there any effects that could possibly remove the degeneracy of the crossing point to reduce the number of WPs? If so, will the net chiral charge number possibly reduce to zero? Is there any symmetry protection that can always keep the total chiral charge of the real materials none zero?

A: In our work, experiments and theory complement with each other and agree well with each other as shown in Fig.2. The good agreement gives us confidence that the results from the DFT calculations are reliable.

On the other hand, the 36 WPs on the generic k points are accidental crossing points which carry a special property differing from normal Weyl nodes. Normal Weyl points appear in pairs with opposite charges in the 3D BZ under same band inversion. Weyl points with opposite charges annihilate when touching each other, but Weyl points with same charges cannot annihilate. However, for the 36 WPs in PtGa with two categories named WP1 and WP2, each category W1 (or W2) has same topological charge lies symmetrically in the 3D BZ under same band inversion as shown in Fig. 4b. It indicates that if the band inversion of W1 (W2) is removed by pressure or another changes, all same-category -12 Weyl charges (+24 Weyl charges) will disappear suddenly. On the other hand, the WP0 degeneracy at the Γ point is always protected by time-reversal symmetry. Thus, when certain types of band inversion disappear, the net charge of all the Weyl points can change from +13 to -11, +25, or 1 in following the formular $1+24\times m-12\times n$, where m and n are either 0 or 1. Therefore, although the net topological charge of the Weyl points inside the BZ (excluding nodal wall) can be different from +13, it is always a non-zero odd number because of the special protected Weyl point at Γ point. We have added more description in the reversed manuscript.

List of main changes:

1. We changed the title.
2. We removed all the sentences “beyond Nielsen-Ninomiya no-go theorem” related to our work in the manuscript. On the other hand, we added more description about the extended Nielsen-Ninomiya no-go theorem.

- 3, We added more information about the photon energy for spin-ARPES measurements.
- 4, We added more explanation about the difference between previous ARPES paper and our work.
- 5, We added more general explanation about Chern number calculations on gapped 2D slices.
- 6, We added more explanation about the topology of net charge of the Weyl points inside one BZ.

REVIEWERS' COMMENTS

Reviewer #1 (Remarks to the Author):

In the revised manuscript, the authors have removed "beyond the Nielsen-Niomiya no-go theorem" from the title and revised the introduction accordingly. Now the new title is more precise.

Regarding the Fermi arc surface states, I agree with the authors, due to the complexity of the band structure and the finite resolution of ARPES, one can not count the number of Fermi arcs to confirm the Chern number of the Weyl point in this material. However, the topological protection is still well-defined between a Weyl plus a nodal wall. It deserves more effect on finding a better material candidate in the future.

Overall, I am satisfied with the authors' response on the revised manuscript. I would recommend the publication of this work.

Reviewer #2 (Remarks to the Author):

I have read the authors' replies. I think that the authors did a great job in answering my questions. I therefore recommend the paper for publication.

Reviewer #3 (Remarks to the Author):

I am satisfied with authors' reply. Therefore, I support its publication in NC.

We would like to thank the Referees for their efforts in reviewing our revised manuscript and for their appreciation of the significance of our work.

~~~~~  
**Reply to Reviewer #1**  
~~~~~

Q: In the revised manuscript, the authors have removed "beyond the Nielsen-Niomiya no-go theorem" from the title and revised the introduction accordingly. Now the new title is more precise.

Regarding the Fermi arc surface states, I agree with the authors, due to the complexity of the band structure and the finite resolution of ARPES, one can not count the number of Fermi arcs to confirm the Chern number of the Weyl point in this material. However, the topological protection is still well-defined between a Weyl plus a nodal wall. It deserves more effect on finding a better material candidate in the future.

Overall, I am satisfied with the authors' response on the revised manuscript. I would recommend the publication of this work.

A: We sincerely thank the Referee for his/her agreement about our response of the concept "Nielsen-Niomiya no-go theorem" and the related modifications.

~~~~~  
**Reply to Reviewer #2**  
~~~~~

Q: I have read the authors' replies. I think that the authors did a great job in answering my questions. I therefore recommend the paper for publication.

A: We thank the Referee for high evaluation of our efforts for the revised manuscript.

~~~~~  
**To Reviewer #3**  
~~~~~

Q: I am satisfied with authors' reply. Therefore, I support its publication in NC.

A: We thank the Referee for being satisfied with our reply.